# HemoDownloader: Open source software utility to extract data from HemoCue HbA1c 501 devices in epidemiological studies of diabetes mellitus

**Martin Rune Hassan Hansen** [1,2]*, **Vivi Schlünssen**[1,2], **Annelli Sandbæk**[3,4]

**1** Research Unit for Environment, Work and Health, Danish Ramazzini Center, Department of Public Health, Aarhus University, Aarhus, Denmark, **2** National Research Centre for the Working Environment, Copenhagen, Denmark, **3** Research Unit for General Practice, Department of Public Health, Aarhus University, Aarhus, Denmark, **4** Steno Diabetes Center Aarhus, Aarhus University Hospital and Central Denmark Region, Aarhus, Denmark

* martinrunehassanhansen@ph.au.dk

**Data Availability Statement:** All relevant data are within the manuscript and its Supporting Information files. The newest version of HemoDownloader is available at GitHub (https://

## Abstract

Diabetes mellitus is a serious disease with increasing global prevalence. Point-of-care analysis of glycated hemoglobin A ($HbA_{1c}$) holds promise as a diagnostic test for diabetes mellitus in epidemiological studies in challenging environments with limited access to centralized biochemical labs. The HemoCue HbA1c 501 device can be used for point-of-care determination of $HbA_{1c}$, but its usability in epidemiological studies is limited by its inability to export results in digital format. We have developed the open source HemoDownloader software to overcome this limitation of the device. HemoDownloader has an easy-to-use graphical user interface and can export data from HemoCue HbA1c 501 to standard spreadsheet file formats. The program has the potential to improve data collection and management in epidemiological studies of diabetes mellitus.

## Introduction

Diabetes mellitus is a group of medical conditions characterized by insulin resistance and/or decreased insulin production, resulting in hyperglycemia [1]. The Global Burden of Disease Study estimated that in 2017, the global prevalence of diabetes mellitus was 476 million, and the diseases caused 1.4 million deaths [2].

The diagnosis of diabetes mellitus is often made based on measurement of fasting plasma glucose (FPG) or an oral glucose tolerance test (OGTT). Both of these tests require patients to fast [3], which can pose a logistical challenge. Glycated hemoglobin A ($HbA_{1c}$) is a compound formed in the bloodstream by a non-enzymatic chemical reaction between glucose and hemoglobin. $HbA_{1c}$ is a measure of the average plasma glucose levels in the last 8–12 weeks [3], and it is approved by the World Health Organization for the diagnosis of diabetes mellitus, provided that appropriate quality control measures are in place, and the patient does not have other medical conditions rendering the results unreliable [3]. $HbA_{1c}$ analysis does not require

github.com/martinrunehassanhansen/
HemoDownloader).

**Funding:** This work was supported by grants from Aarhus University Research Foundation (project number 81231, https://auff.au.dk/en/) and the National Research Centre for the Working Environment (project number 10322, https://nfa.dk/). The funders had no role in study design, data collection and analysis, decision to publish, or preparation of the manuscript.

**Competing interests:** I have read the journal's policy and the authors of this manuscript have the following competing interests: AS reports personal fees from Sanofi Aventis and personal fees from Novo Nordisk DK, outside the submitted work. MRHH and VS have nothing to disclose.

fasting [3], making it an attractive alternative to diagnosis based on FPG and/or OGTT. Furthermore, $HbA_{1c}$ is recommended by the American Diabetes Association as the main measure of glycemic control among patients already diagnosed with diabetes [4].

Epidemiological studies of diabetes mellitus are needed to monitor the burden of the disease and to investigate novel risk factors for its development. Many of the world's diabetic patients live in low- and middle-income countries [2] where conducting epidemiological studies is challenging, especially if biological samples have to be transported long distances for analysis at centralized biochemical labs. Point-of-care devices for the measurement of $HbA_{1c}$ exist and have the potential to simplify data collection for studies of diabetes mellitus in such settings. Our partners and we recently carried out an epidemiological study on glycemic regulation in Uganda [5, 6]. Our main outcome was $HbA_{1c}$, determined by the point-of-care device HemoCue HbA1c 501 (HemoCue AB, Ängelholm, Sweden, www.hemocue.com). No software is currently available for extracting the analysis results from the device memory, so we developed the open source HemoDownloader utility to perform this task, in order to not rely solely on data collectors' manual recording of results.

## Methods

The software was developed for use in and tested during data collection for the "Pesticide Exposure, Asthma and Diabetes in Uganda" (PEXADU) study, which was a short-term observational cohort study on health effects of pesticide exposure among smallholder farmers in the semi-urban Wakiso District of Uganda. We recruited 364 farmers, most of whom underwent three $HbA_{1c}$ tests each from September 2018 to February 2018. Hence, more than 1000 $HbA_{1c}$ analyses were conducted during the study. Study findings have been reported elsewhere [5, 6].

The PEXADU study was conducted in accordance with the Declaration of Helsinki. Study participants gave written informed consent before inclusion and were financially compensated for lost earnings on examination days. The study was approved by the Higher Degrees Research and Ethics Committee at Makerere University School of Public Health (registration number 577) and the Uganda National Council for Science and Technology (registration number HS234ES).

### Software description

The HemoCue HbA1c 501 device saves analysis results to its internal non-volatile memory with a capacity for 200 measurements [7]. Each result is tagged with the date and time of analysis. In addition, a barcode scanner used at the time of analysis can be used to tag each measurement with the IDs of the patient and of the technician performing the analysis [7].

The manufacturer of the HemoCue HbA1c 501 device does not provide software capable of exporting the contents of the internal device memory in digital format. This limits the usability of the device in epidemiological studies with hundreds or thousands of measurements, as results have to be manually recorded. The software that we have created (HemoDownloader) allows a PC to download the entire memory of the HemoCue HbA1c 501 device and save it in a structured file for further processing using standard statistical software. This makes the use of the device for epidemiological studies easier, faster and less error-prone.

HemoDownloader takes advantage of the fact that the HemoCue HbA1c 501 device supports printing all results in memory using a stand-alone thermal printer connected by a serial (RS232) cable [7]. To use HemoDownloader, the PC is connected to the printer port of the HemoCue HbA1c 501 device using a generic RS232 null-modem cable, and the printer function on the HemoCue HbA1c 501 device is activated [7]. HemoDownloader records the binary stream of data intended for the thermal printer, parses the data and saves the dataset on the hard drive of the PC. The software can export to CSV (Comma-Separated Values), TSV (Tab-

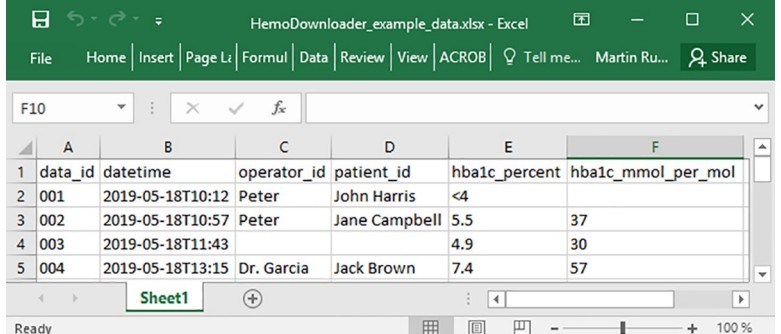

**Fig 1. Example data structure of exported file.** Note: The variables "operator_id" and "patient_id" will be blank if no barcode scanner was used to enter this information before sample analysis. If the HbA$_{1c}$ value in percent is <4% or >14%, the HemoCue HbA1c 501 device does not convert it to mmol/mol, and the latter field is left blank.

Separated Values), XLSX (Microsoft Excel workbook) and XLS (Microsoft Excel 97–2003 workbook) file formats. The resulting file lists the date, time, technician ID, patient ID, and HbA$_{1c}$ analysis result for each measurement, with one line per observation. Fig 1 shows an example Microsoft Excel workbook, demonstrating the structure of the data saved by Hemo-Downloader. For reasons of confidentiality, the data in the example file are fictional.

## Practical use of the software

HemoDownloader is compatible with computers running Microsoft Windows (32 and 64 bit). The compiled versions of the software can be executed directly, do not require installation, and do not depend on any other installed software. Since the HemoDownloader software was developed in the Python 3 programming language, users who want to run the program from source code instead need to install Python 3 (Python Software Foundation, https://www.python.org), as well as the open source Python libraries pyserial, XlsxWriter and xlwt that are used by HemoDownloader. Installation scripts for these third-party libraries are included in S1 File in the online appendix.

Before using the HemoDownloader software, the user has to connect the PC to the Hemo-Cue HbA1c 501 device using a RS232 null modem cable (a standard RS232 serial cable will not work). The user now starts the software and is presented with a simple graphical user interface (Fig 2). Before data can be recorded, the user must specify where to save the resulting file, and

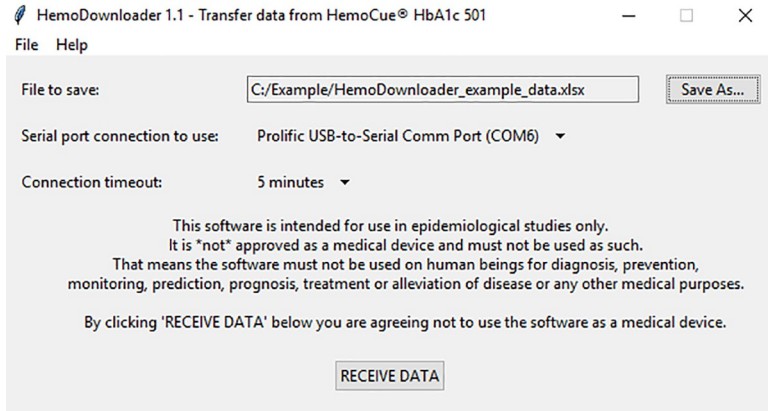

**Fig 2. Main program window.**

which of the PC's serial ports to use. The user clicks the button "RECEIVE DATA" in Hemo-Downloader, and the utility is now ready to receive data.

The printer function of the HemoCue Hba1c 501 device is activated in the normal manner [7]: With the HemoCue HbA1c 501 device in stand-by mode, the user presses the device's MODE button for three seconds, followed by the UP or DOWN buttons to select "Data", and confirming by pressing the MODE button briefly. The HemoCue HbA1c 501 device will now display all results in memory. The user presses the PRINTER button briefly to bring up the print menu. Using UP or DOWN buttons, the user selects "ALL" and confirms by briefly pressing the MODE button. This will initiate the data transfer.

HemoDownloader will automatically close the serial connection when the transfer is complete. The HemoCue HbA1c 501 device only supports data transfer at 9600 baud, but the entire memory of the device can be downloaded in a few minutes.

HemoDownloader understands all date-time formats supported by the HemoCue HbA1c 501 device. The software automatically checks the recorded data for consistency and warns the used in case the data structure is not exactly as expected from the HemoCue HbA1c 501 device.

## Discussion

Measurement of $HbA_{1c}$ instead of FPG or OGTT can simplify the logistics of epidemiological studies on diabetes mellitus, as $HbA_{1c}$ testing does not require fasting [3]. It is an advantage if the analysis can be carried out at the point-of-care, especially in studies in areas with poor access to centralized biochemical labs. The usefulness of the HemoCue HbA1c 501 device for point-of-care analysis of $HbA_{1c}$ in epidemiological studies is hampered by its inability to export data in digital format. HemoDownloader fulfills the need for a utility that can perform this extraction for more efficient data management, and its graphical user interface means that it can be utilized by non-technical users.

In the course of the PEXADU project, the HemoDownloader utility was used for the management of results from more than one thousand $HbA_{1c}$ analyses carried out over the course of several months. $HbA_{1c}$ results were recorded both manually and using HemoDownloader, and any inconsistencies between the recorded values were used to identify and correct bugs in the software. We are therefore confident that the software works as intended, and we hope that HemoDownloader will prove just as useful to other field researchers studying diabetes mellitus.

Please note that HemoDownloader is intended for use in epidemiological studies only. The software is not approved as a medical device and must not be used as such. That means the software must not be used on human beings for diagnosis, prevention, monitoring, prediction, prognosis, treatment or alleviation of disease or any other medical purposes.

## Conclusions

We have presented our utility HemoDownloader that can be used to export $HbA_{1c}$ data recorded by the HemoCue HbA1c 501 device to standard file formats. We believe that Hemo-Downloader has the potential to improve data management in research projects utilizing the HemoCue HbA1c 501 for the study of diabetes mellitus in challenging environments such as low- and middle-income countries.

## Supporting information

**S1 File. Source code for and compiled versions of HemoDownloader 1.1.**
(ZIP)

## Acknowledgments

### License information

HemoDownloader is open source software, licensed under the GNU General Public License (GPL) version 3 [8].

HemoDownloader uses the open source Python libraries pyserial, XlsxWriter and Xlwt. pyserial by Chris Liechti, XlsxWriter by John McNamara, and the parts of xlwt written by John Machin and Manfred Moitzi are all released under BSD licenses. xlwt was forked from the module pyExcelerator developed by Roman V. Kiseliov (roman@kiseliov.ru) that was released under a modified BSD license. Portions of xlwt are based on the module pyXLWriter by Evgeny Filatov and John McNamara, released under the GNU Lesser General Public License version 2.1. The compiled versions of HemoDownloader were created using the program PyInstaller, distributed under a modified GPL license.

HemoCue is a registered trademark of HemoCue AB (Ängelholm, Sweden). This research was not supported by HemoCue AB.

## Author Contributions

**Conceptualization:** Martin Rune Hassan Hansen.

**Data curation:** Martin Rune Hassan Hansen.

**Funding acquisition:** Martin Rune Hassan Hansen, Vivi Schlünssen, Annelli Sandbæk.

**Investigation:** Martin Rune Hassan Hansen.

**Methodology:** Martin Rune Hassan Hansen.

**Software:** Martin Rune Hassan Hansen.

**Supervision:** Vivi Schlünssen, Annelli Sandbæk.

**Validation:** Martin Rune Hassan Hansen.

**Visualization:** Martin Rune Hassan Hansen.

**Writing – original draft:** Martin Rune Hassan Hansen.

**Writing – review & editing:** Martin Rune Hassan Hansen, Vivi Schlünssen, Annelli Sandbæk.

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
