## [Decision Letter · Decision Letter 0]

22 Sep 2020

PONE-D-20-09858

HemoDownloader: Open source software utility to extract data from HemoCue HbA1c 501 devices in epidemiological studies of diabetes mellitus

PLOS ONE

Dear Dr. Hansen,

Thank you for submitting your manuscript to PLOS ONE. After careful consideration, we feel that it has merit but does not fully meet PLOS ONE’s publication criteria as it currently stands. Therefore, we invite you to submit a revised version of the manuscript that addresses the points raised during the review process.

We look forward to receiving your revised manuscript.

Kind regards,

Igor Kanovsky, Ph.D.

Academic Editor

PLOS ONE

Academic Editor's Comment:

The paper does not describe any research and cannot be published as a Research Article.

Nevertheless, PLOSONE can publish another type of papers. This manuscript may be published as a type: **new software and tools**. For details see:

https://journals.plos.org/plosone/s/submission-guidelines#loc-methods-software-databases-and-tools

If you agree with the paper type reclassification, please consider minor changing in the paper as stated in the reviews. Don't pay attention on the research related comments.

Journal Requirements:

Reviewers' comments:

Reviewer's Responses to Questions

**Comments to the Author**

1. Is the manuscript technically sound, and do the data support the conclusions?

Reviewer #1: Partly

Reviewer #2: No

2. Has the statistical analysis been performed appropriately and rigorously? 

Reviewer #1: N/A

Reviewer #2: No

3. Have the authors made all data underlying the findings in their manuscript fully available?

Reviewer #1: No

Reviewer #2: No

4. Is the manuscript presented in an intelligible fashion and written in standard English?

Reviewer #1: Yes

Reviewer #2: Yes

5. Review Comments to the Author

Reviewer #1: This manuscript shows good sound that there is a new way to get data from epidemiological studies only about HbA1c in diabetes melitus. The authors said that this software can be used for analysing thousand samples and it has been implemented in Uganda Research, but the total samples in Uganda research were 370 and it was cohort, which one is true, please explain more detail because the total samples are different. Data were transfered in excel form, this condition was rather difficult to analyse thousands (whole) data in research setting, did it need to integrate with other programme (application)?

I see the software just for transfering data that is in Hemocue and there is not special command in software for analysis. is it true? How did the software analyse 370 or thousand samples? please explain more to know the usefull of the software. How did this software identify the subjects (370 or thousand) if the research was cohort that need many times to perform? Did the author identify data one by one or separate with some instructions in software? The authors may show how data was grouped and analysed with this software (cohort and or cross sectional maybe), also show the benefit and minus of using this software, like in Uganda research that has been done.

The limitation of this software is only for epidemiological research, but may be the author can be improve this software in clinical setting.

Reviewer #2: This is an interesting article that describes the effort to create a software to export results in digital format from point of care device of HbA1c measurement. The development of this software was expected to help the researcher to improve data collection and management in epidemiological study of diabetes.

This study is supposed to be about software development to export stored data from Hemacue system and diabetes.

Critiques

1. The article is too basic discussing information that is more suitable for a software development article than for a research article. This is particularly true for the sections after introduction that describe the software and its practical use.

2. Beyond this much of the article appears to be a series of statements regarding the software and its benefit without presenting evidence to support the statements.

3. The article was written with unusual style in scientific paper in medicine or life science. As a scientific research, the article does not present any method and result to support the authors’ conclusions. Since this article was claimed as research article, the authors should explain the method of the study clearly.

4. Since diabetes is a major health problem, the information regarding diabetes is updated frequently by International Diabetes Federation and WHO. Therefore, the use of reference regarding the incidence and mortality rate of diabetes (line 30) from the report article published more than 5 years ago is not acceptable.

4. The Authors wrote “Epidemiological studies of diabetes mellitus are needed to monitor the spread of the disease and to investigate novel risk factors for its development” (line no.44). I think the use of phrase “the spread of the disease” is not appropriate with the context of diabetes since it is not a transmitted disease.

5. The sentence “Our partners and we recently carried out an epidemiological study on health effects of pesticide exposure among small-scale farmers in the semi-urban Wakiso District of Uganda” (line 49) is not relevant with this article.

6. The authors do not give any strong reason why the project is important. Since the device for measuring HbA1c are available from many companies, the project to provide a software from the only one device from certain company is not relevant if the aim is for epidemiological study.

6. PLOS authors have the option to publish the peer review history of their article (what does this mean?). If published, this will include your full peer review and any attached files.

Reviewer #1: No

Reviewer #2: No

---

## [Author Response · Author response to Decision Letter 0]

3 Oct 2020

Academic Editor's Comment

Comment

The paper does not describe any research and cannot be published as a Research Article. Nevertheless, PLOSONE can publish another type of papers. This manuscript may be published as a type: new software and tools. For details see: https://journals.plos.org/plosone/s/submission-guidelines#loc-methods-software-databases-and-tools

If you agree with the paper type reclassification, please consider minor changing in the paper as stated in the reviews. Don't pay attention on the research related comments.

Response

We agree with the paper type reclassification. Unfortunately, we have been unable to change the type in Editorial Manager ourselves, as the only paper types that can be selected in the system are "Collection Review", "Overview", "Research Article", "Clinical Trial", "Registered Report Protocol" and "Registered Report". We kindly request that you help us change the paper type to “new software and tools”.

Journal Requirements

Comment

Response

We confirm that we have reviewed the PLOS ONE style requirements for the main manuscript, file names, author names and affiliations. Appropriate minor adjustments have been made. Furthermore, we have used the Preflight Analysis and Conversion Engine (PACE) digital diagnostic tool to make sure that our image files live up to PLOS ONE requirements.

Comment

2. Your ethics statement should only appear in the Methods section of your manuscript. If your ethics statement is written in any section besides the Methods, please move it to the Methods section and delete it from any other section. Please ensure that your ethics statement is included in your manuscript, as the ethics statement entered into the online submission form will not be published alongside your manuscript. 

Response

We have created a “Methods” section and moved the ethics statement there, deleting the separate “Ethics” section.

Reviewers' comments

Reviewer #1

Comment

This manuscript shows good sound that there is a new way to get data from epidemiological studies only about HbA1c in diabetes melitus. The authors said that this software can be used for analysing thousand samples and it has been implemented in Uganda Research, but the total samples in Uganda research were 370 and it was cohort, which one is true, please explain more detail because the total samples are different. 

Response

We have now added a “Methods” section, where we have provided more information on the number of participants and samples analyzed.

Comment

Data were transfered in excel form, this condition was rather difficult to analyse thousands (whole) data in research setting, did it need to integrate with other programme (application)?

Response

We used Stata to analyze the extracted data. We have not added this information to the current manuscript, as we do not present any results from the analysis here.

Comment

I see the software just for transfering data that is in Hemocue and there is not special command in software for analysis. is it true?

Response

Before data collection for the study started, we contacted HemoCue (the manufacturer of the HbA1c 501 device), as we wished to purchase data management software for the device. We were informed by the company that such software did not exist.

Comment

How did the software analyse 370 or thousand samples? please explain more to know the usefull of the software.

Response

The HemoCue HbA1c 501 device analyzed more than one thousand samples from 364 individuals. The HemoDownloader software was used to download the data from the analyzer to a PC. HemoDownloader cannot be used for data analysis – this has to be conducted in other software, e.g. Excel, Stata or R.

Comment

How did this software identify the subjects (370 or thousand) if the research was cohort that need many times to perform?

Response

Please see the new “Methods” section as described above.

Comment

Did the author identify data one by one or separate with some instructions in software?

Response

As described in the section “Practical use of the software”, HemoDownloader can be used to simultaneously download all results saved in the memory of the HemoCue HbA1c 501 device (the device has a capacity of 200 results). The software parses to data received to identify individual measurements and saves each measurement as a separate observation in the output file.

Comment

The authors may show how data was grouped and analysed with this software (cohort and or cross sectional maybe), also show the benefit and minus of using this software, like in Uganda research that has been done.

Response

As described above, HemoDownloader was not used to analyze the data, only to download it from the HemoCue HbA1c 501 to a PC.

In the new “Methods” section, we have provided references to two publications in which the results from statistical analyses can be found: The PhD thesis of the first author, and a recent publication in Occupational and Environmental Medicine.

As described in the “Discussion” section, the advantage of using HemoDownloader is that it can make data management easier in epidemiological studies of diabetes mellitus, as researchers using the HemoCue HbA1c 501 do not have to rely on manual data entry. Theoretically, using software to extract the data will lead to a small risk of data corruption due to data transfer errors. However, we implemented a number of data consistency checks in the HemoDownloader software, and we used the software every day for several months of data collection. More than one thousand HbA1c analyses were recorded both manually and using HemoDownloader, and any inconsistencies in data were used to correct software bugs so that the inconsistencies disappeared. We are therefore confident that HemoDownloader functions as intended. To keep the manuscript concise, we have not elaborated in detail on this in the manuscript.

Comment

The limitation of this software is only for epidemiological research, but may be the author can be improve this software in clinical setting.

Response

To allow HemoDownloader to be used in clinical settings, it would have to be approved as a medical device. Unfortunately, we do not have the financial resources to apply for such approval. However, as we are releasing the software as open source, anyone wishing to use the software in a clinical setting is free to apply for this approval him- or herself.

Reviewer #2

Comment

This is an interesting article that describes the effort to create a software to export results in digital format from point of care device of HbA1c measurement. The development of this software was expected to help the researcher to improve data collection and management in epidemiological study of diabetes.

This study is supposed to be about software development to export stored data from Hemacue system and diabetes.

Critiques

1. The article is too basic discussing information that is more suitable for a software development article than for a research article. This is particularly true for the sections after introduction that describe the software and its practical use.

Response

We agree and request that the editor change the manuscript type to “new software and tools”.

Comment

2. Beyond this much of the article appears to be a series of statements regarding the software and its benefit without presenting evidence to support the statements.

Response

As described in the section “Source code availability”, we are releasing the source code for the software in S1 File as well as on GitHub (https://github.com/martinrunehassanhansen/HemoDownloader). While knowledge of the Python programming language is needed to understand the source code, readers can review the source code to confirm how the software works, including the automatic checks for data consistency. Numerous comments have been added in the source code files to make it easier for third parties to review and understand the code.

Comment

3. The article was written with unusual style in scientific paper in medicine or life science. As a scientific research, the article does not present any method and result to support the authors’ conclusions. Since this article was claimed as research article, the authors should explain the method of the study clearly.

Response

We have added a “Methods” section providing basic information on the study for which the software was developed. 

Comment

4. Since diabetes is a major health problem, the information regarding diabetes is updated frequently by International Diabetes Federation and WHO. Therefore, the use of reference regarding the incidence and mortality rate of diabetes (line 30) from the report article published more than 5 years ago is not acceptable.

Response

We have revised this section to use the newest available estimates for the global prevalence and number of deaths caused by diabetes mellitus, based on the Global Burden of Disease Study 2017.

Comment

4. The Authors wrote “Epidemiological studies of diabetes mellitus are needed to monitor the spread of the disease and to investigate novel risk factors for its development” (line no.44). I think the use of phrase “the spread of the disease” is not appropriate with the context of diabetes since it is not a transmitted disease.

Response

We understand how the term “spread” could be misleading and have replaced it with the word “burden”.

Comment

5. The sentence “Our partners and we recently carried out an epidemiological study on health effects of pesticide exposure among small-scale farmers in the semi-urban Wakiso District of Uganda” (line 49) is not relevant with this article.

Response

We have revised this sentence to make it more concise and relevant, and have moved background information on the PEXADU study to the new “Methods” section.

Comment

6. The authors do not give any strong reason why the project is important. Since the device for measuring HbA1c are available from many companies, the project to provide a software from the only one device from certain company is not relevant if the aim is for epidemiological study.

Response

In addition to our own project, the HemoCue HbA1c 501 has been used or evaluated for the assessment of glycemic regulation in at least 12 previous scientific papers. This indicates that software to improve data management from the device could be of use to the scientific community.

While we agree that it would be optimal to have software that could interface with HbA1c analyzers from many different companies, developing such software was outside the scope of our project. However, since we are releasing the HemoDownloader software under the GNU General Public License, anyone will be free to reuse the code and incorporate it in software that can interface with multiple devices.

---

## [Editor Report · Decision Letter 1]

27 Oct 2020

HemoDownloader: Open source software utility to extract data from HemoCue HbA1c 501 devices in epidemiological studies of diabetes mellitus

PONE-D-20-09858R1

Dear Dr. Hansen,

We’re pleased to inform you that your manuscript has been judged scientifically suitable for publication and will be formally accepted for publication once it meets all outstanding technical requirements.

Kind regards,

Igor Kanovsky, Ph.D.

Academic Editor

PLOS ONE
---

## [Editor Report · Acceptance letter]

3 Nov 2020

PONE-D-20-09858R1 

HemoDownloader: Open source software utility to extract data from HemoCue HbA1c 501 devices in epidemiological studies of diabetes mellitus 

Dear Dr. Hansen:

I'm pleased to inform you that your manuscript has been deemed suitable for publication in PLOS ONE. Congratulations! Your manuscript is now with our production department. 

Kind regards, 

on behalf of

Professor Igor Kanovsky 

Academic Editor

PLOS ONE